# Acoustic and Thermal Properties of Particleboards Made from Mulberry Wood (*Morus alba* L.) Pruning Residues

Manuel Ferrandez-Villena *, Antonio Ferrandez-Garcia, Teresa Garcia-Ortuño and Maria Teresa Ferrandez-Garcia

Department of Engineering, Universidad Miguel Hernández, 03300 Orihuela, Spain;
antonio.ferrandezg@umh.es (A.F.-G.); tgarcia@umh.es (T.G.-O.); mt.ferrandez@umh.es (M.T.F.-G.)
* Correspondence: m.ferrandez@umh.es; Tel.: +34-966-749-716

**Abstract:** The aim of this study was to determine the acoustic and thermal properties of particleboards made from mulberry wood pruning waste using urea formaldehyde resin (UF) as a binder. The investigation focused on the evaluation of the thermal conductivity and the acoustic absorption of the boards and the assessment of their feasibility for use in the construction sector. The mean thermal conductivity values of the particleboards (0.065–0.068 W/mK) were lower than those obtained in wood and similar to those in cork panels. The samples were tested with frequencies from 50 to 6300 Hz. In all cases, the results allowed us to conclude that they were better sound absorbers than commercial wood and plywood panels of the same average density for low frequencies, and with similar values for medium and high frequencies. The mechanical results reached the minimum requirement to be considered as boards for general use and, specifically with particles from 0.25 to 1.00 mm, for furniture according to European standards. The particle size of the particleboards was the variable that influenced all the acoustic properties, but did not affect the thermal conductivity. The experimental results indicated that the thermal and acoustic properties of these particleboards were promising for their application in commercial uses.

**Keywords:** acoustic absorption; insulation; valorization; composite; waste

## 1. Introduction

Currently, there are specialized materials of mineral origin (rock wool, fiberglass, etc.) on the market with good acoustic and thermal properties and synthetic origin (polyurethane foam, polystyrene, etc.). These materials have the drawback of having a high energy consumption during their manufacture and are not biodegradable. For this reason, the use of renewable and ecological resources is increasing and will be the common practice for insulation in building construction.

The most common organic material used as a sound absorber is wood in the form of panels or particleboards. In general, lignocellulosic materials are porous and good sound absorbers, with acoustic insulation properties in a wide range of frequencies. The fact that vegetable fibers can be cheaper, lighter, and more environmentally friendly justifies their research as an alternative to synthetic fibers [1].

As substitutes for acoustic materials, several investigations have been carried out with particleboards that have used different plant residues, including rice stalks (*Oryza sativa* L.) [2], coconut fibers (*Cocos* L.) [3], bamboo (*Dendrocalamus asper* (Schult.) Backer) [4], jute (*Corchorus* L.) with latex [5], oil palm (*Elaeis guineensis* Jacq.) residues [6], date palm (*Phoenix dactylifera* L.) fibers [7], canary palm (*Phoenix canariensis* hort. Ex Chabaud) residues [8], and giant reed (*Arundo donax* L.) [9]. Likewise, other studies have been carried out using date palm fibers [10], hemp (*Cannabis sativa* L.) [11], sisal (*Agave sisalana* Perrine) [12], and canary palm residues [8] as thermal insulators.

Mulberry (*Morus alba* L.) trees were widely cultivated in the Levante area to feed silkworms. They are currently grown in Spain as ornamental and shade trees in urban

landscaping. Mulberry trees are pruned at least once a year. These prunings produce one- or two-year-old softwood that is normally disposed of in landfill. The current trend is to improve the sustainable use of resources, so it is intended to take advantage of waste and obtain new products.

The specific objective of this study was to determine the thermal and acoustic properties of boards made with mulberry pruning residue particles as substitutes for insulating materials and to evaluate their industrial application.

The advantages of using this residue would be the valorization of a waste without use into a product, lower the accumulation of waste in landfills, and a similar industrial process to wood but with lower energy consumption in drying particles and grinding. The possible disadvantages would be the collection of pruning once a year and its volume depending on an uneven territorial distribution.

## 2. Materials and Methods

The materials used in this investigation were one-year-old mulberry pruning branches collected from the Orihuela campus of Miguel Hernández University. The branches were dried outside for 12 months. The adhesive used was 8% wt. (based on the weight of the particles) urea formaldehyde resin (UF) class E1 (adhesive that is used in the manufacture of wood-based panels with E1 class formaldehyde emissions) with a solid content concentration of 65% and reactivity of 3–4 h. As a hardener, ammonium nitrate was used at a concentration of 0.4% wt. The ratio of particles to adhesive mass is shown in Table 1.

**Table 1.** Ratio of particles to adhesive mass of the experimental panels.

| Particles (g) | UF Solid Content (g) | Water (g) | Ammonium Nitrate (g) |
|:---:|:---:|:---:|:---:|
| 100 | 5.2 | 2.8 | 0.4 |

The particles were obtained in a blade mill and then were classified according to their size with a vibrating sieve into three categories: from 0.25 to 1.00 mm, 1.00 to 2.00 mm, and 2.00 to 4.00 mm. The moisture content on the dry basis of the particles is shown in Table 2.

**Table 2.** Humidity of the particles used in the manufacturing of the particleboards.

| Particle Size (mm) | Moisture Content (%) | Standard Deviation |
|:---:|:---:|:---:|
| 0.25–1.00 | 8.03 | 0.54 |
| 1.00–2.00 | 8.25 | 0.06 |
| 2.00–4.00 | 8.59 | 0.42 |

The particleboards were manufactured in a hot plate press at a temperature of 120 °C, a pressure of 2.5 MPa, and for 5 min. Four boards with approximate dimensions of 600 mm × 400 mm × 10 mm were made with each particle size used, generating a total of 12 boards.

To determine the mechanical properties of the mulberry particleboards, specimens of each board were obtained [13] and their average density [14], modulus of rupture (MOR) and the modulus of elasticity (MOE) [15], and internal bonding strength (IB) [16], were evaluated as indicated by the European standards.

The average density and the mechanical tests were determined by a universal testing machine (model IB600, Imal, S.R.L., Modena, Italy). The bending test was carried out with six samples from each board (three in a longitudinal direction and three in a transversal direction) measuring 250 × 50 × 10 mm, at a constant velocity of 5 mm/min. The IB test was performed with three samples from each board measuring 50 × 50 × 10 mm, taken from the outer and inner parts of the board, using a constant velocity of 2 mm/min. Before testing, the samples were placed in a JP Selecta refrigerated cabinet (model Medilow-L, Barcelona, Spain) for 24 h at a temperature of 20 °C and relative humidity of 65%.

Thermal conductivity and resistance were determined by the stored hot plate method and the heat flow meter method [17] and were carried out in a heat flow meter (NETZSCH Instruments Inc., Burlington, MA, USA). For this test, a specimen of each board with dimensions of 300 mm × 300 mm was used (Figure 1).

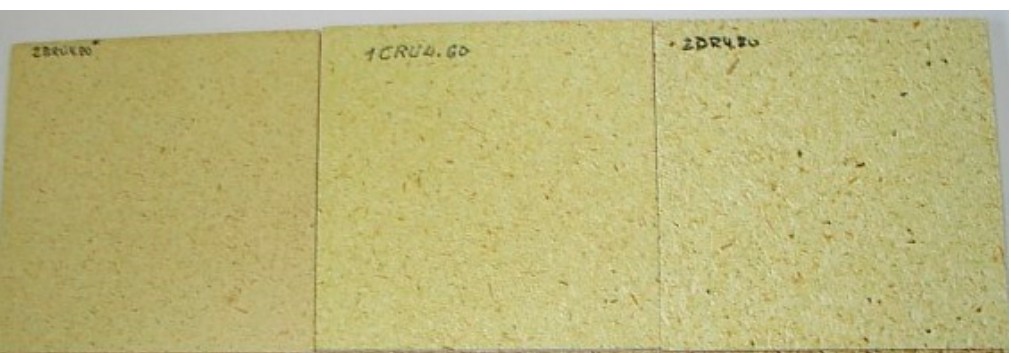

**Figure 1.** Experimental panel samples for thermal conductivity testing.

The method followed to determine the acoustic absorption coefficient of a material ($\alpha$) under normal incidence is based on the acoustic impedance tube. This test method uses an impedance tube, two microphone positions, and a digital signal analysis system [18]. This technique requires a previous test correction procedure to minimize the differences in the amplitude and phase characteristics between the two microphones. To carry out the tests, the Acupro Spectronics impedance tube (Spectronics C., Lexington, KY, USA) was used, with a range of frequencies between 50 and 6300 Hz and using 3 specimens from each board.

The standard deviation was obtained from the mean values of the tests and analysis of variance (ANOVA) was performed. Statistical analyses were performed with SPSS v. 28.0. (IBM, Chicago, IL, USA).

Their properties were determined according to the European standards established for wood particleboards [19].

## 3. Results and Discussion

### 3.1. Physical and Mechanical Properties

In order to proceed with the laboratory tests, the specimens extracted from the manufactured particleboards were kept in a controlled atmosphere at a temperature of 20 °C and a relative humidity of 65%. The mean values obtained from the tests are shown in Table 3.

**Table 3.** Characteristics of the experimental panels.

| Particle Size (mm) | Average Density (kg/m³) | MOR (N/mm²) | MOE (N/mm²) | IB(N/mm²) | Thermal Conductivity (W/mK) |
|---|---|---|---|---|---|
| 0.25–1.00 | 848.75 (30.50) | 17.98 (0.56) | 1974 (36) | 1.43 (0.15) | 0.06722 (0.003) |
| 1.00–2.00 | 843.06 (12.63) | 14.84 (0.97) | 1466 (23) | 1.54 (0.20) | 0.06983 (0.013) |
| 2.00–4.00 | 807.13 (48.54) | 11.50 (0.98) | 1351 (108) | 2.30 (0.21) | 0.06536 (0.002) |

(..) Standard deviation.

The average density of the boards ranged between 848.75 and 807.13 kg/m³; they were classified as medium-density panels. The MOR of the boards made from particles from 0.25 to 1 mm reached a value of 17.98 N/mm². However, it decreased when the particle size increased further. MOE values varied from 1351 to 1974 N/mm². These results followed the same trend as MOR values. In contrast, IB results improved as the particle size increased.

This may be because Mulberry wood is a hard wood, with a dense grain and strongly intertwined fibers. Hence, bigger particles have more internal cohesion than smaller particles bonded by the adhesive.

Branches of the mulberry trees obtained by pruning have been used by other authors to manufacture different types of boards. Yu et al. [20] produced scrimber boards based on mulberry branches with granulometry of 1 to 5 mm thick and 200–250 mm long by a process at 135 °C and by adding 8% phenol formaldehyde (PF) through a different manufacturing process. They obtained MOR and MOE values of 80.1 $N/mm^2$ of 9820 $N/mm^2$ respectively, which are higher than the values obtained in this study. Mulberry particleboards made from FP may have better mechanical properties than those made with UF. However, it has the disadvantages due to its difficult handling and the multistep process followed.

Using branches of red pine (*Pinus brutia*) without bark with 8% UF and a press temperature of 150 °C, Sahin and Arslan [21] obtained particleboards with a MOR value of 4.66 $N/mm^2$ and a IB value of 0.98 $N/mm^2$, both lower than in the present study. They also noted that particle size influenced the mechanical behavior of the panels.

Particleboards made of Greek fir wood with UF resulted in MOR and MOE values of 30.0 $N/mm^2$ and 4330 $N/mm^2$, respectively [22]. These results are higher than the values reported in this work. However, the authors also noted that the mechanical properties decreased when adding particles from branches of hard wood (such as Mulberry wood).

The mechanical results indicated that the three types of experimental particleboards manufactured reached the minimum requirement to be considered as boards for general use (P1); specifically, with particles from 0.25 to 1.00 mm, they can be used for furniture (P2) according to European standards [23].

### 3.2. Thermal Properties

The experimental particleboards had a mean thermal conductivity ($\lambda$) of 0.067 W/mK. The results are shown in Table 3. This is a very good result compared with the values of various commercial particleboards and others obtained by different authors using other plant residues, as presented in Table 4. From this comparison, we could conclude that with mulberry wood pruning, better thermal behavior was observed than for commercial wood particleboards, date palm, and hemp panels; similar results to cork and sisal; and lower thermal performance than with canary and washingtonia palms.

**Table 4.** Thermal conductivity results obtained by different authors.

| Reference | Material | Thermal Conductivity (W/mK) |
|---|---|---|
| Present work | Mulberry wood prunnings | 0.067 |
| [24] | Cork panel | 0.065 |
| [24] | Commercial wood particleboard | 0.180 |
| [8] | Canary palm | 0.059 |
| [25] | Canary palm + cement | 0.057 |
| [25] | Washingtonia palm + cement | 0.059 |
| [10] | Date palm | 0.083 |
| [11] | Hemp | 0.111 |
| [12] | Sisal | 0.070 |

### 3.3. Acoustic Properties

Figure 2 shows the values obtained for the acoustic absorption coefficient ($\alpha$) of the tests performed on the three specimens of each type of experimental particleboard. It could be observed that there were major differences between the boards based on the size of the particles. With the three types of boards, a high value is obtained for very low frequencies (at 50 Hz, the absorption coefficient was 0.45); it decreased for medium frequencies to low values, and again increased but at a different rate for each particleboard type.

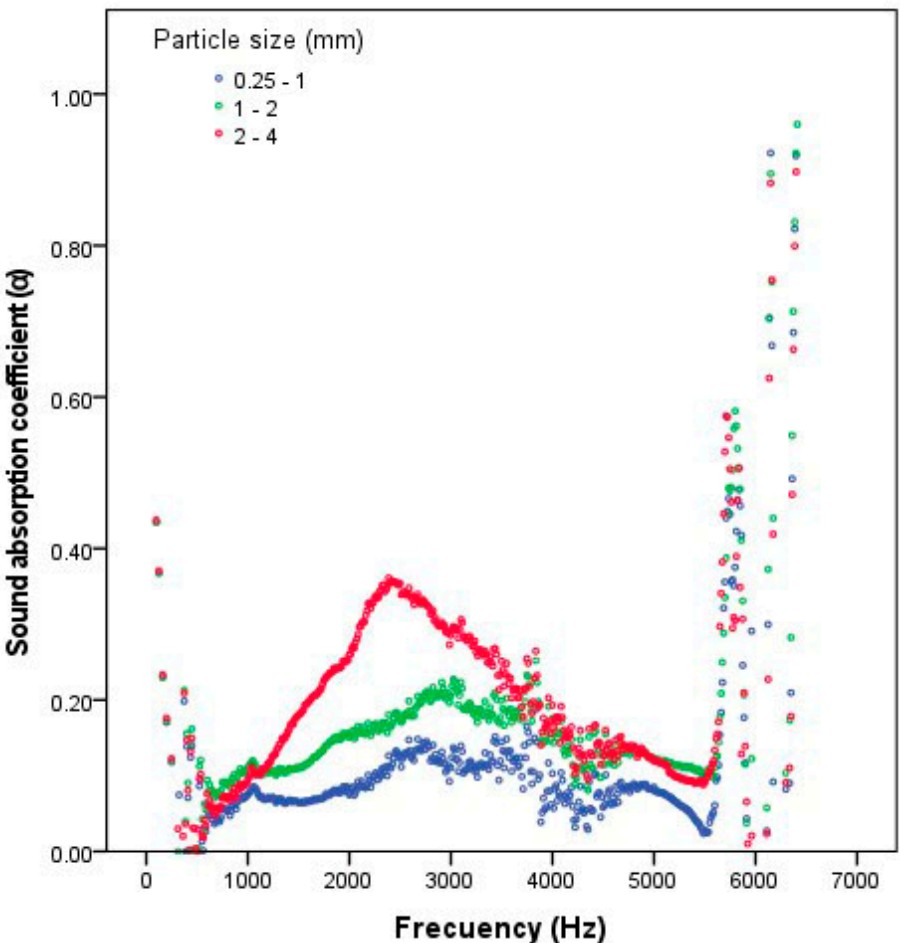

**Figure 2.** Absorption coefficient results according to the particle size of the panels.

The graphs of the different types of panels indicated that there are significant differences among the boards, especially in the bands from 1250 to 4000 Hz. This may be due to the porosity of the boards due to the gaps between the particles. With larger particles (2.00 to 4.00 mm), a higher porosity is observed in the particleboards.

Boards are classified in building construction [26] according to their acoustic coefficient: class D boards have values between 0.30 to 0.5, class E between 0.15 to 0.25, and boards with lower acoustic coefficient values are unclassified. In general, mulberry boards with 2 to 4 mm particles are classified as class E acoustic absorbent boards, except in the frequency range from 500 to 1000 Hz.

Panels made from canary palm [8] and giant reed [9] displayed the same behavior, with a decrease in the sound absorption coefficient for medium frequencies and increases in the low and high frequency ranges. The sound absorption coefficient in these two studies was influenced by the particle size. In contrast to rice strow [2], particle size did not influence the sound absorption coefficient, but did affect the average density. In the present work, with similar densities, higher acoustic absorption values have been observed with bigger particles.

Table 5 shows the acoustic absorption coefficients observed according to the central frequencies of normalized octave bands. These frequencies are most common in architectural acoustics and in most of the works and studies consulted, and facilitate the subsequent comparison of the results with the values obtained with other materials of the same average density commonly used in construction.

**Table 5.** Acoustic absorption coefficient according to frequency.

| Material | Particle Size (mm) | Frequency (Hz) | | | | | |
|---|---|---|---|---|---|---|---|
| | | 125 | 250 | 500 | 1000 | 2000 | 4000 |
| 0.25–1.00 | 0.25–1.00 | 0.368 | 0.118 | 0.037 | 0.073 | 0.079 | 0.082 |
| 1.00–2.00 | 1.00–2.00 | 0.368 | 0.120 | 0.049 | 0.111 | 0.155 | 0.157 |
| 2.00–4.00 | 2.00–4.00 | 0.371 | 0.122 | 0.049 | 0.116 | 0.227 | 0.157 |
| Wood [27] | – | 0.150 | 0.110 | 0.100 | 0.070 | 0.060 | 0.070 |
| Plywood [27] | – | 0.280 | 0.220 | 0.170 | 0.090 | 0.100 | 0.110 |

Transmission loss (TL) is a parameter that depends on frequency. It is measured in decibels (dB) and its value indicates the decrease in sound intensity that is reduced by when passing through a material. Figure 3 shows the average values of TL obtained in the test of three samples of each type of board based on the central frequencies of normalized octave bands.

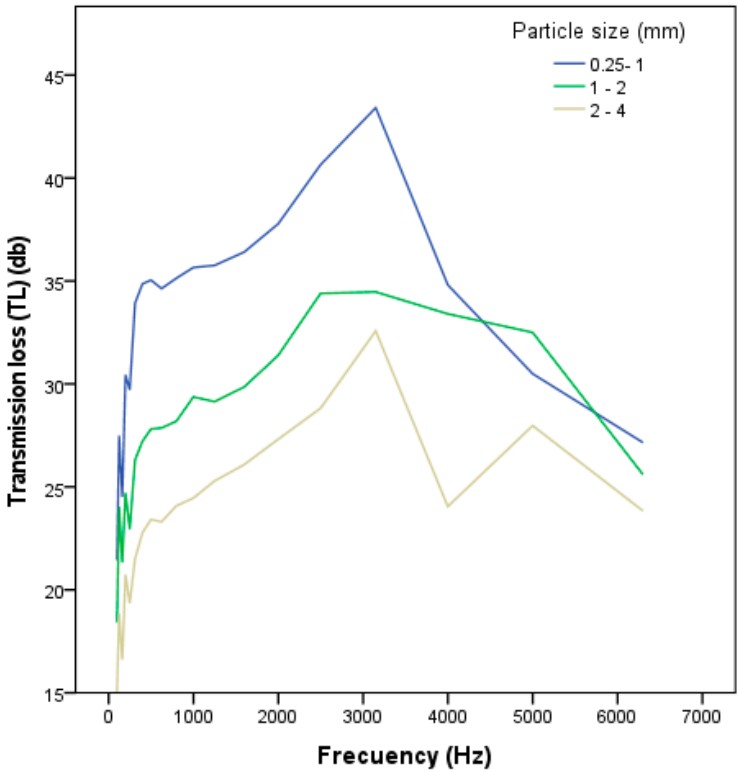

**Figure 3.** Transmission loss (TL) results according to the particle size of the panels.

TL values were higher for boards with a smaller particle size, with an average of 35.5 dB. Boards with a larger particle size had lower TL values. Therefore, it is possible to conclude that this parameter is influenced by the particle size. Again, this could be conditioned by the gaps between its particles.

### 3.4. Statistical Analysis

Table 6 shows that the analysis of variance (ANOVA) of the properties obtained from the boards indicates that there was a dependency ($p$-Value < 0.05) between MOR, MOE, and IB values and the particle size used in manufacturing of the board, whereas the average density and thermal conductivity were not influenced by this parameter.

**Table 6.** ANOVA of the results of the tests.

| Factor | Properties | Sum of Squares | d.f. | Half Quadratic | F | *p*-Value |
|---|---|---|---|---|---|---|
| | Density (kg/m$^3$) | 4,075,034 | 2 | 2037.517 | 1.770 | 0.225 |
| | MOR (N/mm$^2$) | 83,815 | 2 | 41.908 | 14.102 | 0.002 |
| Particle size | MOE (N/mm$^2$) | 1,207,600.503 | 2 | 603,800.251 | 9.920 | 0.005 |
| | IB (N/mm$^2$) | 1.062 | 2 | 0.531 | 6.537 | 0.018 |
| | Thermal C. (W/mK) | 0.000 | 2 | 0.000 | 0.316 | 0.737 |

d.f.: degrees of freedom. F: Fisher–Snedecor distribution.

## 4. Conclusions

Particleboards made from mulberry wood pruning residues with good mechanical properties have been successfully manufactured and could be classified as P1 (for general uses) according to EN 312.

When manufactured with a particle size from 0.25 to 1.00 mm, the particleboards could be classified as type P2; therefore, they could be used in furniture production.

All experimental boards tested had mechanical properties superior (P1 and P2) to those required for boards intended for acoustic and thermal insulation.

Mulberry particleboards may be good thermal insulator panels in comparison to other commercial particleboards made from lignocellulosic materials.

With particle sizes from 2 to 4 mm, the particleboards are adequate for use as insulation panels for sound absorption.

**Author Contributions:** Idea and methodology: M.F.-V. and A.F.-G. Experiments: T.G.-O. and A.F.-G. Resources: M.F.-V. Statistics: T.G.-O. and M.F.-V. Project administration: M.T.F.-G. Supervision: M.T.F.-G. Writing: M.F.-V. Review: M.T.F.-G. All authors have read and agreed to the published version of the manuscript.

**Funding:** This research was funded thanks to Agreement No. 4/20 between the company Aitana, Actividades de Construcciones y Servicios, S.L., and Universidad Miguel Hernandez, Elche.

**Institutional Review Board Statement:** Not applicable.

**Informed Consent Statement:** Not applicable.

**Data Availability Statement:** The data presented in this study are available within the article.

**Acknowledgments:** The authors would like to thank the company Aitana, Actividades de Construcciones y Servicios, S.L. for its support by signing Agreement No. 4/20 with Universidad Miguel Hernández, Elche on 20 December 2019.

**Conflicts of Interest:** The authors declare no conflict of interest.

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
