# Peer review of "Acoustic and Thermal Properties of Particleboards Made from Mulberry Wood (Morus alba L.) Pruning Residues"

_agronomy, doi:10.3390/agronomy12081803_

Round 1

Reviewer 1 Report

Manuscript is interesting. It is a good supplement to the knowledge of the influence of the type of raw material and particle size on the properties of the manufactured particleboards. However, I have a few comments:

1. In table 1, in the fourth column, MOE should be instead of MOR (this is evidenced by the values given)

2. It is interesting that the authors observe the increase in the value of IB with the increase in the size of the particles used to produce the particleboard. An attempt should be made to explain this phenomenon.

3. In the chapter Results and discussion on the mechanical and physical properties of the manufactured particle boards, there is no discussion and reference of the obtained values to the literature.

4. Figure 2 and table 3 present the same information - this is a repetition. From table 3, graph 2 can be recreate.

5. Conclusion: Mulberry particleboards are good thermal insulators in comparison to other commercial particleboards. It's too general. The authors did not make the boards from conventional raw materials, nor did they test commercial boards, therefore this conclusion does not seem to be justified. At least so worded.

6. There is no significance level given in the statistical analysis - it should be given.

7. Please provide the recipe of the adhesive mass (e.g. mass share of resin to hardener)

Reviewer 2 Report

The manuscript is dealing with the measurement of acoustic, mechanical, and thermal properties of particleboards made from mulberry wood waste. In abstract, the acoustic and thermal properties are mentioned, but in the paper also measurement of modulus of elasticity, modulus of rupture, and internal bounding strength are listed. This has to be added to the abstract.

My remarks and questions (referring to the line-numbers):

36 – 39: I suggest to add Latin names of trees, it will be more understandable.

 61 and 92: right will be “120 °C” and “20 °C”

 61: instead of “Mpa” right will be “MPa”

 63 – 66: The determination of MOR, MOE and IB should be described in more detail.

 64: In whole manuscript, do not use “density”. Particleboard is porous material; in this case must be used quantity “bulk density” or “average density”.

Table 1: Particle size has to be in mm. Instead of density, right will be bulk density. There MOR is twice, second abbreviation has to be MOE.

110: “table 2” right will be “Table 2”

Table 2: Instead of p-Value, right will be Thermal conductivity (W/mK).

Figure 2: What does (a) means on the axis of acoustic absorption coefficient? Probably, authors wanted to write (a). This quantity is dimensionless, and right will be (-).

Reviewer 3 Report

Review of the manuscript ID: agronomy-1774851

Acoustic and thermal properties of particleboards made from mulberry wood
(Morus Alba L.) pruning residues

The manuscript submitted for review deals with the acoustic and thermal insulation properties of particleboards produced from particles extracted from the wood of the mulberry tree Morus Alba L. The authors additionally determined the mechanical properties of the experimental boards, which allowed them to be classified as appropriate particleboards. The topic of this work is in line with current trends in the search for alternative raw materials (both woody and non-woody) in the manufacturing process of wood-based boards. Despite this, however, the work raises a number of concerns, which are presented below.

Specific comments:

1.  Abstract, line 10 - use notation: "...urea formaldehyde resin (UF) as a binder." The word "resin" is missing. A similar comment applies to the text in section 2 Materials and Methods, line 54 - "urea formaldehyde resin".

2.  Keywords - what do the authors mean by "valorization"? I think this keyword does not fit the context and content of the manuscript.

3.     In my opinion, the Introduction of the article was too laconic. The concise content therein regarding alternative raw materials essentially amounts to a listing of works in this area. I think that due to the subject of the manuscript, this part of the work should be more elaborated, e.g., what are the advantages and disadvantages of the proposed solutions? What are the properties of boards derived from these raw materials in relation to traditional boards made of raw wood, etc.?

4.     Materials and Methods:

a.     line 54 - the E1 class is the class of wood-based panels, not the resin, so it should be stated that this is an adhesive that is used in the manufacture of wood-based panels with E1 class formaldehyde emissions.

b.     line 55 - the reactivity of the resin, which according to the Authors is 3-4 h, is questionable. Usually, the reactivity of adhesive resins is determined by gel time at 100°C, activation energy or by DSC thermograms.  Since the unit of given reactivity is hours, what tests were used to determine this reactivity? Is it reactivity or is it lifetime?

c.   incorrectly stated amount and concentration of hardener solution. Please correct this. Is the 0.4% wt. the concentration of the hardener solution or the amount added per weight of solution or dry weight of the UF resin?

d.     what was the moisture content of the particles from the mulberry wood?

e.     line 61- please correct the units of pressure to MPa.

5.     Results and Discussion:

a.     subsection 3.1. and 3.2. - lack of any discussion of the obtained research results, i.e. discussing them in the perspective of previous studies, lack of reference to the work of other authors. These fragments of the manuscript require rewriting and supplementing. There is also no explanation of the way in which the obtained strength values were formed depending on the size of wood particles. While this is obvious information to wood-based panel specialists, it is not understandable to a potential reader who is not an expert in this field of science.

b.     I request that the Authors justify the exceptionally high IB values.

c.      subsection 3.1 - Thermal conductivity values are given in Table 1, but the description of these values is in subsection 3.2.  Please rewrite this section of the manuscript and move the values to subsection 3.2.

d.     subsection 3.2., Table 2 - in the heading of column 3, the p-value is given instead of the thermal conductivity coefficient. The thermal conductivity value for commercial particleboard is questionable. The Authors have provided a value of 1,800 (without units), but this is an incorrect value. Please indicate on which page of reference 21, it was given, as I did not find such information. The typical particle board has a thermal conductivity of 0.13-0.16 W/mK.

e.     incorrect numbering of paragraphs under the title Statistical analysis and Conclusions

f.       there is inconsistency between the sentence on page 3, line 101 "In contrast, IB results improved with larger particle size." and the assignment on page 5, lines 160-161. First, the Authors claim that increasing the particle size improves IB, but in the statistical analysis they say that it does not matter. The validity of the statistical analysis and the conclusion drawn is questionable since the IB values for particles of 0.25-1.0 mm and 2.0-4.0 mm fractions differ by as much as 60%.

6.     Conclusion - the conclusion needs to be clarified. The sentence on page 6, lines 170-171, is a conclusion that goes too far and misleads the potential reader. It should be clarified or deleted.

7.     The entire manuscript should be linguistically corrected, there are many grammatical errors.

Recommendation

The manuscript was prepared without due professional and editorial care. It requires thorough revision. Despite its poor quality, I believe it has potential, and therefore recommend it for major revision.  

Round 2

Reviewer 1 Report

1. in a sentence: The bending test was carried out with six samples from each board (three in a longitudinal direction and three in a transversal direction). what does longitudinal i transversal direction mean? The article presents research on particleboard. What, according to the authors, mean these directions?

2. line 139 is "Using braches" should be "Using branches"

3. Still, chart 3 shows the data contained in table 5. Of course, there are other materials in the table, but the chart is a repetition of the data on the material tested and does not have any additional information than can be found in table 5.

4. In conclusions, when referring to European standards, there should be an indication of a specific standard that defines classes P1, P2 ....

Reviewer 2 Report

The authors corrected the manuscript according to most of my suggestions. But some mistakes remained and there are also new ones.

My remarks and questions (referring to the line-numbers):

67: right will be “Table 1”

72 and Table 2: The quantity “humidity” is used in case of gaseous substances, especially for air. For solid substances, it is necessary to use the quantity “moisture content”. It is still necessary to distinguish between moisture content wet basis and moisture content dry basis. Moisture content dry basis (moisture content db) is mostly used for wood and its products. This must be corrected.

75 - 76, 117, 134, 140: the symbol for degree (°C) should not be underlined

90: there must be a space between the number and the unit, right will be “20 °C”

139: "Pinius brutia" must be in Italics

140: “Turgut Sahin et al [21]”, right will be “Sahin and Arslan [21]”; In References Sahin H. T.; Arslan M. B.

Figure 2: Instead of (a) on the axis of acoustic absorption coefficient, right will be (a).

281: there should not be a parenthesis at the beginning of the citation

Reviewer 3 Report

Review of the manuscript ID:  agronomy-1774851

Acoustic and thermal properties of particleboards made from mulberry wood (Morus Alba

The Authors made changes to the peer-reviewed manuscript, but not all of the Reviewer's comments were addressed, and some of the corrections made still raise serious concerns

Specific comments:

1.     The revised manuscript still raises concerns about the reactivity of the UF resin. In my opinion, it is not reactivity versus lifetime. If it is reactivity, which is 3-4 h, then how do the Authors explain the short pressing time of 5 min? These parameters, which are closely related, should work together.2.     Still my doubt is the concentration and amount of hardener. Can an ammonium nitrate solution with such a low concentration and such a small amount lower the pH of the UF resin enough to initiate its crosslinking process under pressing conditions. I have serious doubts about this. Usually, this kind of hardener is used at 2% by weight relative to the dry weight of the resin and at a concentration of 10-20%. These doubts also arise from the Authors' assertions that this resin has a "reactivity" (?) of 3-4 h, so we conclude that it is low-reactive.

3.     In Table 2, the moisture content of the vortices should be designated as MC (moisture content). RH refers to the relative humidity of the air. Also, in the table header and line 72 instead of humidity there should be moisture content. I suggest the moisture content of the particles be reported as one average value for all fractions. I leave the decisions to the Authors.

4.     In the discussion of the results on page 4, lines 134-138, reference was made to the work of Yu et al. (2015), which deals with the use of mulberry branches of scrimber plastics, not particleboard as stated by the Authors. This is a completely different wood-based materials. After all, Yu et al. used wood strands of 200-250 mm in length and 1-5 mm in diameter, which is quite a different form of raw material than the Authors. This passage should definitely be rewritten, as it gives incorrect information. The cited paper studied the effect of density and resin content on properties such as MOR, MOE and shear strength, without IB.

5.     There is still no explanation for the very high IB values noted by the Authors.  These are values (especially for chips of 2-4 mm in size) that even exceed the requirements of the standards by several times, as well as values usually obtained for particleboards glued with UF resin with a relatively low degree of gluing. Hence, this issue requires clarification.

My recommendation so far remains unchanged - major revision.
